# Abrasion Effect on Heating Performance of Carbon Nanotube/Epoxy Composites

**DOI:** 10.3390/nano15050337

**Published:** 2025-02-21

**Authors:** Byung-Wook Kim, Seung-Jun Lee, Sung-Hwan Jang, Huiming Yin

**Affiliations:** 1Department of Civil Engineering and Engineering Mechanics, Columbia University, 500 W 120th Street, New York, NY 10027, USA; bk2825@columbia.edu; 2Department of Civil and Environmental Engineering, Hanyang University, 222 Wangsimni-ro, Seongdong-gu, Seoul 04763, Republic of Korea; sj5523@hanyang.ac.kr; 3Department of Civil and Environmental Engineering, Hanyang University, 55 Hanyangdaehak-ro, Sangnok-gu, Ansan 15588, Republic of Korea

**Keywords:** carbon nanotube, composites, Joule heat, abrasion

## Abstract

The effects of abrasion on the heating performance of carbon nanotube (CNT)/epoxy composites were investigated in terms of Joule’s heat, convective heat, and radiative heat under moderate-to-severe and localized abrasive conditions. While the overall heating behavior was characterized by the heating rate and the curvature of the transient response, a numerical solution of the heat equation was used to quantify convective and radiative heat transfers, incorporating the specific heat of each component, the convective heat transfer coefficient, and the Biot number. CNT reinforcement significantly improved wear resistance at a CNT concentration of 0.31 vol. %, but the presence of micro-voids led to a slight increase in wear rate with additional CNT inclusion. Using an equivalent circuit model, local and severe abrasion scenarios were analyzed to determine the variation in electrical conductivity with temperature at different degrees of abrasion, indicating the impact of scattering effects. This analysis provides valuable insights for estimating both wear resistance and the heating performance of self-heated surface materials, with potential applications in future space technologies.

## 1. Introduction

Incorporating carbon-based fillers into a polymeric matrix enhances the resulting composites with a broad spectrum of electrical and mechanical properties. Due to their high aspect ratio and conductivity, a small amount of the fillers enables the construction of percolation networks, demonstrating tunable composites from insulators to conductors [1,2], and they also reinforce other mechanical properties, such as tensile strength and Young’s modulus [3,4,5]. In addition to the tailorable properties of the composites, the solution-based synthesis method is effective in providing scalable composites with customizable dimensions and thicknesses of molds [6]. Such unique characteristics prompt the consideration of various applications by adjusting electrical resistance with carbon filler content and selecting appropriate polymers for each purpose. Using the flexibility of polymers with appreciable thermal and chemical resistance [5,7] when carbon fillers are dispersed in a polymeric matrix, flexible heaters can be developed to generate Joule heat using electrical current through fillers for various applications: carbon nanotube (CNT)/epoxy composites have been considered for de-icing because epoxy resin can be integrated into structures such as bridges, terraces, roofs, and helicopter rotor blades [8,9,10,11]; flexible carbon fiber composites can also be used for de-icing curved aircraft wings, as these are compatible with passenger aircraft such as the Airbus A350 XWB and Boeing 787, which primarily consist of carbon fiber-reinforced polymer composites [12,13,14]; and flexible CNT composites have been evaluated to provide thermal comfort for passengers in vehicle seats [15,16].

In particular, motivated by the Artemis project, which seeks to protect astronauts from harsh environments and to improve the capability of mission tasks to the Moon and potential advancements for Mars [17], materials used in extravehicular activity (EVA) systems must be lightweight as space exploration costs thousands of dollars per pound [18]. While other lightweight fillers, such as halloysite and boron nitride nanotubes, can provide flame retardancy and radiation shielding, respectively [19,20], CNT-based composites can offer tunable electrical conductivity, enabling both cosmic radiation shielding and temperature control through Joule heating [21,22]. Recently, although the material design is beneficial for thermoset materials for on-site repair in cold regions with less power consumption and without additional curing equipment [23,24,25], heat generation in tandem with abrasion problems has been reported during extravehicular activities (EVAs) [26]. For example, while the temperature of the lunar surface drops to 25 K without solar illumination, spacesuits, vehicles, and terrestrial habitats are exposed to micrometeoroid collisions as well as abrasion by lunar dust, which is much more severe than the weather on Earth [27]. As the electrostatic potential of the lunar dust varies from positive on the sun-facing side to negative on the opposite side depending on photoelectrons produced by solar wind and UV rays, an EVA object attracts dust due to the potential difference and subsequently experiences severe abrasion due to dust sharpness, which has been shaped without any wind or water erosion [28,29]. Indeed, NASA reported that the abrasion of EVA suits during an 8 h lunar-surface activity was more severe than 100 h usage abrasion of the training suits, meaning that the suits were limited to 75 h use [30]. Previously, lunar dust abrasion has been simulated by Taber wheels of up to 8000 cycles [30,31], and the abrasion of polymer composites has been investigated with a focus on the coefficient of friction and wear rate, namely an enhanced coefficient of friction and thermal conductivity due to interconnected fillers [32], lower coefficient of friction due to reduced hardness [33], wear rate related with surface porosity [34], and orientational effect of fillers on the coefficient of friction and wear rate [35].

However, previous studies remain insufficient for space applications based on further demands of required thermal and abrasive properties. As summarized in Table 1 [23,27], EVA suits have undergone more than 2500 h exposure [25], showing non-uniform abrasion to different parts [23,27,30]. Therefore, considering the flame retardancy, heating, and curing of the CNT/epoxy composite [5,7,23], this work investigates its heating performance under severe and local abrasions that may occur in application environments.

## 2. Materials and Methods

### 2.1. Fabrication of CNT/Epoxy Composites

Chemical vapor deposition (CVD)-grown CNTs were purchased to provide fillers with a length of 5–20 µm, diameter of 10–20 nm, and a density ($\rho_{f}$) of $2.3\;{\mathrm{g/cm}}^{3}$ from NanoLab, Inc. (Waltham, MA, USA), and Smooth-On ${\mathrm{EpoxAcast}}^{\mathrm{TM}}$ 690 was purchased from Easy Composites Ltd. (Staffordshire, UK) for epoxy matrix with a density ($\rho_{m}$) of $1.10\;{\mathrm{g/cm}}^{3}$. Thus, each volume fraction of CNTs (ϕ) in the CNT/epoxy composites was obtained as $\phi ={\left[1+\left(m_{m}\rho_{f}\right)/\left(m_{f}\rho_{m}\right)\right]}^{-1}$, where $m_{m}$ and $m_{f}$ are the measured epoxy matrix mass and CNT filler mass, respectively.

For dispersed CNT fillers within a polymeric matrix, the CNT/epoxy composite films were fabricated through the procedure as shown in Figure 1 [24,36]: (i) once the epoxy precursor (${\mathrm{EpoxAcast}}^{\mathrm{TM}}$ 690: Part A) of 20 g was dissolved with acetone of 50 g in a beaker using a spatula, the desired amount of CNTs (0–2.44 vol. %) was added to the epoxy precursor/acetone solution; (ii) an ultrasonicator (Q700CA, Qsonica LLC, Newtown, CT, USA) was operated to disperse the CNTs in the solution in a pulsed mode of 90% amplitude for 30 min, while the epoxy precursor/acetone/CNTs mixture was surrounded with ice to prevent the evaporation of acetone; (iii) a 60 °C hot plate was used to evaporate the solvent for 24 h, and 6 g of curing agent (${\mathrm{EpoxAcast}}^{\mathrm{TM}}$ 690: Part B) was added to the mixture using a 3-roll mill (TR 50M, Trilos, San Ramon, CA, USA), followed by molding; (iv) air bubbles in the films were removed in a vacuum chamber for 1 h.

### 2.2. Characterizations

Once the CNT/epoxy composites were prepared through the fabrication procedure, in order to minimize the contact resistance between the materials and tip probes for electrical measurements, RS PRO silver conductive paints (RS 186-3600) and ${\mathrm{3M}}^{\mathrm{TM}}$ copper foil tapes were applied to make electrodes at both ends of each specimen. The electrical resistance (*R*) of each specimen was obtained from the slope (1/R) of current–voltage (*I*–*V*) curves by applying *V* from −10 V to +10 V using multimeters depending on *R*: Tektronix Keithley 2700 (Beaverton, OR, USA) for R<10 GΩ and Tektronix Keithley 2450 (Beaverton, OR, USA) for R>10 GΩ [9], so that its electrical conductivity (σ) was calculated as σ=L/RWd, where *L* is the length between the electrodes, *W* is the width, and *d* is the thickness of each specimen [37]. For the temperature coefficient of resistance (α), *R* was measured at different temperatures using an environmental chamber (Lab Companion TE3-KE, Daejeon, Republic of Korea). Laser-flash analysis (NETZSCH LFA 447, Burlington, MA, USA) was used to determine thermal conductivity once both sides of the composites were coated with graphite spraying (Sprayon LU204, Cleveland, OH, USA) for better signal-to-noise ratio by eliminating reflection and enhancing absorption and emission [38].

In addition to the conductivity measurements, a DC power supply (Tektronix 2260B-800-1, Beaverton, OR, USA) was used to generate Joule heat through the specimens with L=80 mm, W=20 mm, and d=5 mm at room temperature. During the heating, their surface-temperature distributions were monitored using a thermal infrared (IR) camera (FLIR A655sc, Wilsonville, OR, USA), ensuring the temperature remained within the IR detector range [9].

According to ASTM D 4060, the standard abrasion test method, abrasion resistance was evaluated for the 5 mm thick disk specimens with 95 mm diameter using a Taber abrasion tester (Standard Solution CT–5100) as shown in Figure 2a. Under 1 kg load of CS–17 abrasive wheel with a rotational speed of 80 rpm, the weight loss (Δm) was determined as the weight difference between before and after 2000 abrasion cycles to within 0.1 mg accuracy during 10,000 cycles of 1727.786 m sliding distance, and the wheel surface was scrubbed to remove any contaminants and to maintain consistent abrasion every 2000 cycles. Thus, the specific wear rate ($W_{s}$) was obtained as $W_{s}=\Delta m/\rho LF$, where the sliding the density is ρ, distance is *L*, and the applied load is *F* [31,35]. Furthermore, a 16 mm wide hand-sander was used to severely and locally abrade the specimens, and the reduced thickness of each abraded area was measured. Different abrasion experimental setups have been reported [39,40].

The morphology of the specimens was examined using a scanning electron microscope (TESCAN MIRA3 FE-SEM, Brno, Czech Republic) at 15 kV on crosss-sections of the abraded areas after Pt coating with a sputter (QUORUM Q150 TS, Laughton, UK). The images were processed to evaluate porosity without any interaction or modification of the specimens [34]. This can also be obtained by gas adsorption or using water [41,42].

## 3. Results and Discussion

### 3.1. Electrical Percolation and Heating Performance

The average of multiple σ measurements was plotted with its error bars of the standard deviation at the corresponding ϕ of CNTs in each specimen in Figure 3a. As a sharp increase in σ was observed from 0.125 to 0.62 vol. % of ϕ, when ϕ is close to the electrical percolation threshold ($\phi_{c}$) [43], the power law was applied to analyze the percolation behavior of σ as follows [44,45,46]:(1)$$\sigma ∼\left\{\begin{array}{c} {\left(\phi_{c}-\phi \right)}^{-s},\phi <\phi_{c}\\ {\left(\phi -\phi_{c}\right)}^{t},\phi >\phi_{c} \end{array}\right.$$
where *s* and *t* are the critical exponents for $\phi <\phi_{c}$ and $\phi >\phi_{c}$, respectively. Thus, using the nonlinear least-square regression, $\phi_{c}$ of 0.25 vol. % was experimentally obtained with the universal and material independent *t* of 2.0 [47]. The effective conductivity at $\phi_{c}$ is related to $R_{f}^{u}R_{m}^{1-u}$, where $R_{f}$ is the resistance of the filler, $R_{m}$ is the resistance of the matrix, and u=s/(s+t) [48,49]. Likewise, since thermal conductivity (κ) was analyzed with the power relation, the same $\phi_{c}$ of 0.25 vol. % was obtained for thermal conduction as plotted in Figure 3a [46,49]. Since both σ and κ began to saturate at ϕ=0.62 vol. %, it is unlikely that further enhancements will be observed with the addition of CNTs beyond ϕ=2.44 vol. %. Moreover, although the aspect ratio (A.R.) of the purchased CNTs initially ranged from 250 to 2000 with the provided length and diameter, it is necessary to estimate the effective A.R. for resultant CNT fillers in the polymeric matrix through the fabrication procedure, since the dispersion energy by ultrasonication reduces agglomerate size as well as break the CNTs, resulting in less length or A.R. [50,51]:(2)$$\phi_{c}=\frac{C}{\frac{4\pi }{3}+2\pi \left(A.R.\right)+\frac{\pi }{2}{\left(A.R.\right)}^{2}}\left[\frac{\pi }{6}+\frac{\pi }{4}\left(A.R.\right)\right]$$
where the constant *C* is 1.4 for thin rods. As $\phi_{c}$ was plotted with A.R. in Figure 3b, the CNTs with A.R. of 285 constructed the percolation networks at $\phi_{c}=0.25$ vol. % after the material fabrication. After constructing percolation networks, as electrons can still hop across the interphase layer due to van der Waals interactions between CNTs [52], the thickness of the interphase layer would be(3)$$t_{int}=\frac{1}{2}{\left(\frac{\phi_{c}}{\phi }\right)}^{1/3}d_{c}$$
where $d_{c}$ is the maximum distance of the electron tunneling between close adjoining CNTs [53]. Considering $d_{c}$ = 1.8 nm [52,53,54,55], as $t_{int}$ = 0.4–0.8 nm with the obtained $\phi_{c}$ of 0.25 vol. % for 0.31≤ϕ≤2.44 vol. %, the both percolation networks and electron hopping contribute to the effective electrical conductivity of the composites for $\phi >\phi_{c}$ [52,56].

In addition, the percolating networks of the CNT fillers are the conducting pathways in which electrical current flows to generate Joule heat ($P_{Joule}$), also referred to as resistive or ohmic heat, which describes the process of heat generation [57]. Figure 4 shows the heating performance in terms of time and temperature distribution for the CNT/epoxy composites with different CNT concentrations from 0.31 to 2.44 vol. % for σ>0.01 S/m, which allows the investigation of their thermal properties with sufficient current flow given the low voltage difference for general heating elements [58]. As plotted in Figure 4a, the average temperature (*T*) of each specimen was monitored with elapsed time (*t*) by an IR camera. *T* rapidly increased at the beginning and then became saturated after around 600 s. Figure 4b illustrates thermal images when the specimens reached their steady states. The uniform distribution of temperature indicates the well-dispersed CNT fillers within the polymeric matrix. While the average and maximum are almost the same, about a 5 K difference was monitored between maximum and minimum temperatures of 2.44 vol. %, as the majority of the central area represents both the average and maximum temperatures in Figure 4b. Furthermore, when an electric current flows through a conductive specimen, $P_{Joule}$ increases *T* with *t* while convective heat ($Q_{conv}$) and radiative heat ($Q_{rad}$) are transferred to ambient environments [59,60,61]. Assuming that such energy conversion is conserved without any additional loss and that the internal temperature gradient is negligible due to simultaneous heating over the entire material through the percolating networks [57], the following heat equation can be written as follows:(4)$$\begin{array}{cc} \left(c_{p,f}m_{f}+c_{p,m}m_{m}\right)\frac{dT}{dt} & =P_{Joule}-Q_{conv}-Q_{rad};\\ P_{Joule} & =\frac{V^{2}}{R_{0}\left[1+\alpha \left(T-T_{0}\right)\right]};\\ Q_{conv} & =hA(T-T_{0});\\ Q_{rad} & =\epsilon \alpha_{SB}A\left(T^{4}-T_{0}^{4}\right) \end{array}$$
where $c_{p,f}$ is the filler specific heat, $m_{f}$ is the filler mass, $c_{p,m}$ is the matrix specific heat, $m_{m}$ is the matrix mass, $R_{0}$ is the resistance at ambient temperature $T_{0}$, α is the temperature coefficient of resistance, *h* is the convective heat transfer coefficient, *A* is the area, ϵ is the emissivity, and $\alpha_{SB}$ is the Stefan–Boltzmann constant. Here, both the radiation coefficients of ϵ and $\alpha_{SB}$ are independent of environmental conditions, whereas *h* can be varied depending on space atmospheric environments, as shown in Table 1. Please note that Equation (Equation 4) was numerically solved in this study since it is impossible to find an analytical solution for general $T>T_{0}$, whereas previous investigations have applied the Taylor expansion to $Q_{rad}:T^{4}-T_{0}^{4}\approx 4T_{0}^{3}\left(T-T_{0}\right)$ for small temperature difference, $T\approx T_{0}$ [59,60]. Therefore, when $P_{Joule}=V^{2}/R$ at V=20 V, using experimental values of *R*, $m_{f}$, $m_{m}$, and *T* with *t* in Figure 4a for each specimen, $c_{p,f}=347$ J/kgK and $c_{p,m}\;=$ 1140 J/kgK were obtained using the nonlinear least-square method and the numerical solution of Equation (Equation 4), while the ranges of $c_{p,CNT}$ = 133–496 J/kgK [61,62,63] and $c_{p,epoxy}$ = 1025–1600 J/kgK [64,65,66,67] have been reported in the literature for the specific heats of CNT and epoxy, respectively, at room temperature. While there exist other methods to measure specific temperatures by preparing additional samples, the nonlinear least-square method offers a more straightforward approach to obtaining the specific heat from the same composite sample used to demonstrate heating performance. This eliminates the need to account for sample-to-sample variation, making the proposed method suitable for this study.

Given those results, when it reaches the steady state, there is no temperature change with time: as the left-hand side of Equation (Equation 3) becomes zero, $P_{Joule,steady},Q_{conv,steady}$, and $Q_{rad,steady}$ are plotted as a function of ϕ in Figure 4c, and higher σ with ϕ will generate higher $P_{Joule}$ and *T*. More specifically, as shown in the inset of Figure 4c, the ratio of $Q_{rad,steady}/P_{Joule,steady}$ increases with higher ϕ and $Q_{conv,steady}/P_{Joule,steady}$ decreases, because of the radiation term of $T^{4}$ in Equation (Equation 4), although $Q_{rad,steady}/P_{Joule,steady}$ was smaller than $Q_{conv,steady}/P_{Joule,steady}$ at ϕ=0.31 vol. %. Figure 4d shows the temperature coefficient of resistance (α) with ϕ. The negative temperature coefficient (NTC) behavior and initial Joule heat reduce *R* and consequently contribute to higher Joule heat in Equation (Equation 4). Such NTC behavior would be considered to be the contribution of electron tunneling, whose transmission probability is inversely proportional to the distance between CNTs in relatively low or intermediate temperatures [68,69]. Furthermore, we have obtained the Biot number $Bi=hd/\kappa_{exp}$, where *d* is the characteristic length or specimen thickness, as shown in Figure 4d with the experimentally obtained $\kappa_{exp}$ in Figure 3a and *h* from Equation (Equation 4), which is 1.6–$14.3\;{\mathrm{W/m}}^{2}$K for natural convection [60,70]. Please note that Bi is the lowest at ϕ=2.44 vol. %, indicating the most effective heat conduction relative to convection through the composites.

### 3.2. Abrasion Characterizations of CNT/Epoxy Composites

To characterize the degree of abrasion, the weight loss averages of the CNT/epoxy composites were primarily obtained in every 2000 abrasive cycles up to 10,000 cycles, as shown in Figure 5a. Overall, the addition of CNTs significantly enhanced abrasion resistance since the pure epoxy had a maximum weight loss of 321 mg after 10,000 abrasion cycles, whereas the 0.31 CNT vol. % showed a minimum weight loss (Δm) of 113 mg, or about 1/3 of the maximum. Such a difference in weight loss is attributed to Archard’s principle, in which hardness is inversely proportional to weight loss. Dispersed CNTs that interact with the epoxy matrix enhance resistance to deformation, leading to increased hardness [71]. Thus, the increased hardness of the composites by the CNT incorporation decreases the plowing effect and reduces the weight loss [72]. Additionally, once the epoxy was preferentially abraded in the CNT/epoxy composites due to local hardness difference, CNTs would be released to form a CNT layer on the surface, giving rise to self-lubrication or a lower coefficient of friction [35,73].

On the other hand, while the specific wear rate ($W_{s}$) significantly decreased with ϕ from 0 to 0.31 vol. %, it started to continuously increase with ϕ from 0.31 to 2.44 vol. %. When observing $W_{s}$ at 10,000 cycles with different CNT concentrations in Figure 5b, this is even more obvious: after showing the highest $W_{s}$ of $0.0266\;{\mathrm{mm}}^{3}$/Nm at 0 vol. %, the $W_{s}$ showed the lowest of 0.0108 to $0.0137\;{\mathrm{mm}}^{3}$/Nm at 0.31 and 2.44 vol. %, respectively. Such behavior occurs because of the inhomogeneous properties of composite materials when the CNT concentration exceeds the percolation threshold of $\phi >\phi_{c}$. Even though the percolating networks were formed under the uniform distribution of CNTs, the further addition of CNTs induces local agglomeration, which leads to pores inside, and the interface between the CNT and the matrix is weakened [26,74]. Indeed, Figure 5c–e shows SEM images of the abraded surfaces for ϕ of 0, 1.23, and 2.44 vol. %, respectively. As a bigger and larger number of pores were observed, porosities of 0.34% and 1.93% were obtained by image-processing for ϕ = 1.23 and 2.44 vol. %, respectively [34]. Recently, an optimum porous microstructure of fiber-reinforced composites for wear resistance has been reported based on the relationship between porosity and wear rate. Furthermore, CNT inclusion slightly decreases hardness due to porosity and CNT agglomeration [32,34], which corresponds to this work. Moreover, as it has been considered that higher porosity contributes to reducing κ [34], Figure 3a shows that κ remains almost the same although ϕ increases from 1.23 to 2.44 vol. %. Therefore, such excess CNT fillers not only induce more pores but also lead to greater weight loss of the composites during abrasion due to the reduced hardness as well as limited κ.

### 3.3. Abrasion Effect on Heating Performance of the CNT/Epoxy Composites

In addition to the general heating and abrasive behaviors of the composites, the local and severe abrasion effects on the heating performance of the composites were investigated with a degree of abrasion. Based on the previous observation of heating and wear performance, ϕ=2.44 vol. % was chosen for further abrasion investigations due to the lowest Bi, while $W_{s}$ remained relatively similar to different ϕ. Subsequently, for even more prolonged or severe abrasions, the degree of abrasion was defined as the reduced thickness ratio ($d_{a}/d$), where *d* is the initial thickness and $d_{a}$ is the abraded thickness. As the thickness in the middle of the composite was reduced by abrasion, the temperature of the abraded area was monitored and analyzed with the numerical model of Equation (Equation 4) as shown in Figure 6a. Compared to the remaining unabraded area or the composite of $d_{a}/d$ = 0, the abraded area heated up faster with higher temperature at the same applied voltage of 20 V across the specimen. In particular, when $d_{a}/d\geq 0.6$, the temperature of the abraded area rapidly raised with the significant increase of slope, contributing to reaching a steady state within 200 s. For each plot in Figure 6a, therefore, such heating performance was quantified by the heating rate, as the average slope of temperature increased over the elapsed time, and the maximum curvature (*k*) is calculated as(5)$$k=\left|\frac{d^{2}T}{dt^{2}}\right|{\left|1+{\left(\frac{dT}{dt}\right)}^{2}\right|}^{-\frac{3}{2}}.$$

While the heating rate reflects how fast the temperature rises, the maximum curvature serves as an indicator of how efficiently each steady state is reached. Interestingly, the heating rate and *k* showed similar behavior in Figure 6b, as they continuously increased with $d_{a}/d$ in the beginning and rapidly increased after $d_{a}/d$ = 0.6, while the highest values of both were at $d_{a}/d$ = 0.95. This is not simply because of reduced mass or heat capacity of the abraded section, since both the corresponding resistance and voltage increased with less thickness at the same time. Thus, considering the environment when the composite is locally abraded when a constant voltage is applied across it, it is necessary to analyze thermal behavior using an appropriate model. On the other hand, whereas the temperature of the abraded section at the steady state ($T_{steady,a}$) does not either continuously increase or decrease with $d_{a}/d$, it is highest when $d_{a}/d$ = 0.6. As shown in Figure 6c, since the Joule heat at the abraded section ($P_{Joule,a}$) was dominant over convective heat at the abraded section ($Q_{conv,a}$) or radiative heat at the abraded section ($Q_{rad,a}$), the highest $P_{Joule,a}$ corresponds to the highest $T_{a}$ at $d_{a}/d$ = 0.6, while $P_{Joule}$ primarily determined overall behavior of $T_{a}$. Moreover, using the measured temperature of the abraded section ($T_{steady,a}$) and temperature of the unabraded section ($T_{steady,u}$) at steady state, as shown in Figure 7a, Figure 6d indicates that $P_{Joule,a}$ was more concentrated than the rest at the higher degree of abrasion, as both the temperature ratio and Joule heat ratio between the abraded and unabraded sections increased with $d_{a}/d$.

Therefore, it is necessary to analyze $P_{Joule,a}$ in terms of material properties, including the unabraded section, since they are electrically connected to mutually influence the corresponding Joule heat:(6)$$P_{Joule,a}=\frac{V_{a}^{2}}{R_{a}}={\left(\frac{R_{a}}{R_{a}+\Sigma_{i}R_{u,i}}V\right)}^{2}\frac{1}{R_{a}}={\left(\frac{V}{R}\right)}^{2}\frac{L_{a}}{W\left(d-d_{a}\right)\sigma_{a}}$$
where $V_{a}$ is the voltage, $R_{a}$ is the resistance, $L_{a}$ is the length, *W* is the width, and $\sigma_{a}$ is the conductivity for the abraded section. $\Sigma_{i}R_{u,i}$ is the resistance summation of the unabraded sections as $R=R_{a}+\Sigma_{i}R_{u,i}$ according to the schematic and the equivalent circuit diagram in Figure 7a,b. Given the applied voltage (*V*) to the composite, the total resistance (*R*) increased with less thickness ($d-d_{a}$) or a higher degree of abrasion ($d/d_{a}$) so that the Joule heat of the entire composite decreased. However, since the voltage across the abraded section with $R_{a}$ is $V_{a}=\left(R_{a}/R\right)V$, $V_{a}$ increases with higher $R_{a}$ at the same *V*, and the corresponding Joule heat of the abraded section has the following relationship: According to Equation (Equation 5), showing the quadratic response of $P_{Joule,a}$ to external variable *V*, one can estimate higher $P_{Joule,a}$ at higher $d_{a}$ with constant $\sigma_{a}$, but Figure 7c shows that $\sigma_{a}$ was not consistent with $d_{a}/d$ and it almost doubled at $d_{a}/d$ = 0.95 compared to others. However, increased $R_{a}$ is expected with a smaller thickness of $d-d_{a}$. Although it has been previously reported that the electrical conductivity of graphene oxide/polymer composites significantly dropped with abrasive wear because of the deformation of the conducting network [29], $\sigma_{a}$ decreased by only about half with $d_{a}/d$ in this work. The cross-sectional SEM images show CNT distributions of $d_{a}/d$ = 0 (upper left) and at $d_{a}/d$ = 0.95 (upper right) in Figure 7b. Therefore, the CNT/epoxy composites still have robust conducting networks against severe abrasion. Furthermore, while the highest $P_{Joule,a}$ contributed to the highest $T_{steady,a}$ when $d_{a}/d$ = 0 in Figure 6c, $\sigma_{a}$ generally tends to be the opposite of $T_{steady,a}$ with $d_{a}/d$ as shown in Figure 7c, due to the scattering effect. At higher $T_{steady,a}$, more electrons would be scattered by phonons, resulting in reduced mobility and, consequently, lower $\sigma_{a}$ [75,76]. Therefore, as we showed that the composites with local and severe abrasions can still be modeled and used for heating applications, one can consider that it is possible to practically predict both the abrasion degree and heating performance of the CNT composites under moderate-to-severe environments.

## 4. Conclusions

This study investigated the heating performance of CNT composites with a degree of abrasion for space applications and modeled the electro-thermo-mechanical behaviors in terms of the equivalent circuit. Highly conductive composites have been successfully prepared by dispersing CNTs in epoxy matrices, and the variations of electrical and thermal conductivities showed the percolation threshold at ϕ = 0.25 vol. % with the power-law relationship. Time-dependent heat generation was analyzed to obtain $c_{p}$ = 347 and 1140 J/kgK for CNT and epoxy, respectively, by the numerical solution of the heat equation, including Joule heat with convective and radiative heat losses. While the highest wear resistance was found at ϕ = 0.31 vol. %, the wear rate gradually increased with higher ϕ due to induced micro-voids. Further local and severe abrasion effects on heating performance were also analyzed to find the relationship between temperature and Joule heat based on the heat equation and the circuit model. Thus, this investigation provides the foundation for the design and analysis of CNT-polymer composites for self-heated surface materials in space applications, which are subjected to external loads or interactions with other environmental conditions, which are considered to be future work, like At. O exposure in low-Earth orbits [77]. By including other effects, the resistance-based method can still be useful for monitoring composite status in aircraft and for designing composite heaters, including multifunctional sensor and heater applications [13,16], and an optimal CNT concentration can be chosen to balance the required heat generation and weight loss by abrasion.

## Figures and Tables

**Figure 1 nanomaterials-15-00337-f001:**
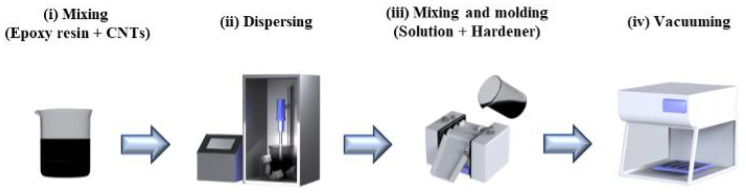
Fabrication procedure of the CNT/epoxy composites: (**i**) epoxy precursor/acetone/CNTs mixture; (**ii**) dispersing CNTs by ultrasonication; (**iii**) mixing curing agent in a 3-roll mill and molding specimens; (**iv**) removing air bubbles in a vacuum chamber.

**Figure 2 nanomaterials-15-00337-f002:**
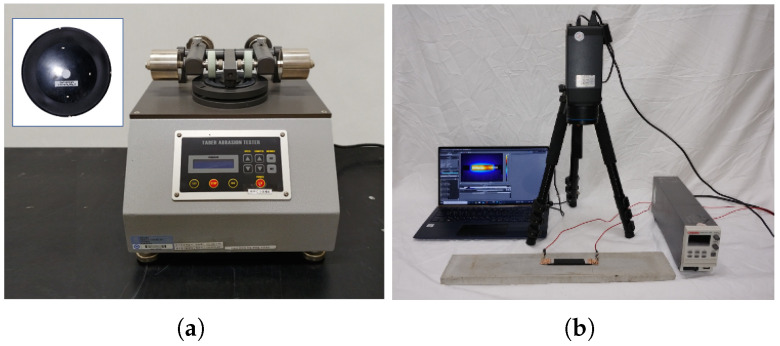
Images of characterization methods: (**a**) abrasion test method with a Taber tester and (inset) a specimen of 9.5 cm diameter and 5 mm thickness, (**b**) Joule-heating method with a DC power supply and an IR camera.

**Figure 3 nanomaterials-15-00337-f003:**
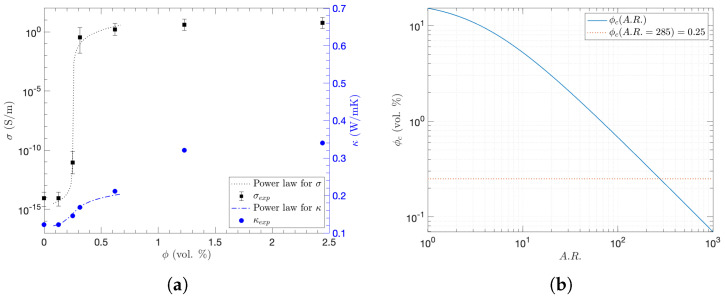
Percolation behavior in the CNT/epoxy composites: (**a**) measured electrical conductivity (σ) and thermal conductivity (κ) as a function of CNT filler concentration (ϕ) with the power-law relationship, (**b**) percolation threshold ($\phi_{c}$) as a function of CNT aspect ratio (A.R.).

**Figure 4 nanomaterials-15-00337-f004:**
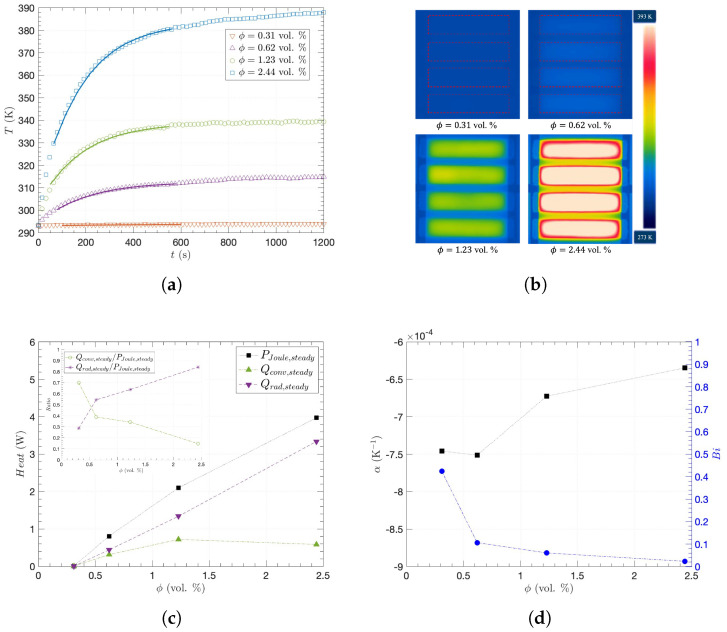
Heating performance of the CNT/epoxy composites with different CNT concentrations: (**a**) the surface temperatures changing in time with the heat-equation model (solid lines), (**b**) the thermal images taken by an IR camera at steady state, (**c**) convective heat ($Q_{conv,steady}$) and radiative heat ($Q_{rad,steady}$) with the heat ratios (inset) to Joule heat ($P_{Joule,steady}$) at steady state, (**d**) temperature coefficient of resistance (α) and Biot number (Bi) as a function of CNT filler concentration (ϕ).

**Figure 5 nanomaterials-15-00337-f005:**
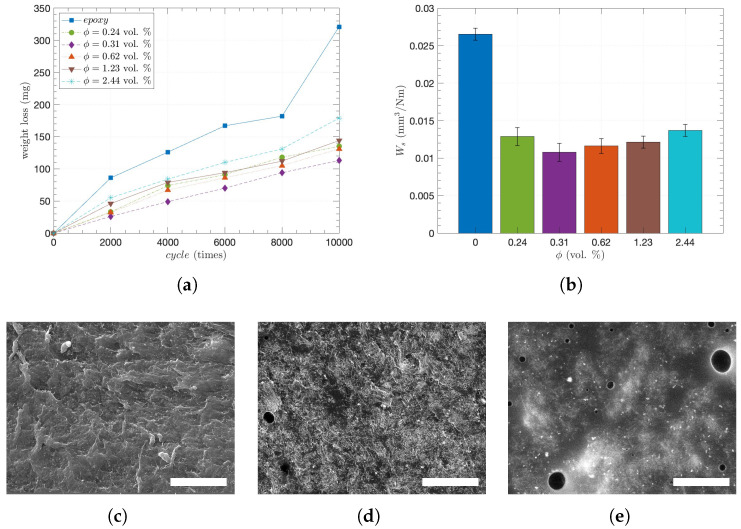
Abrasion characteristics of the CNT/epoxy composites: (**a**) weight loss (Δm) in every 2000 cycles band (**b**) wear rate at 10,000 cycles with the different CNT concentrations, SEM images with 10μm scale bars of the abraded surface with the CNT concentrations of (**c**) 0, (**d**) 1.23, and (**e**) 2.44 vol. %, respectively.

**Figure 6 nanomaterials-15-00337-f006:**
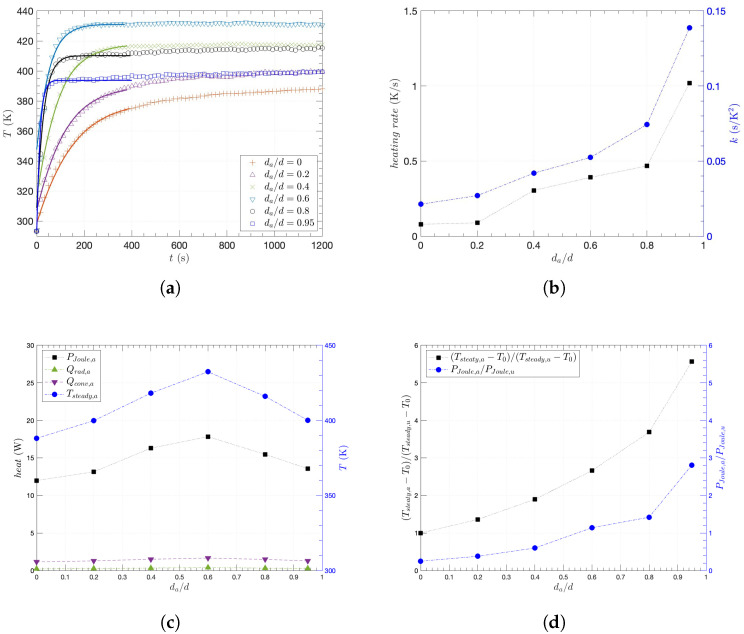
Heating characteristics of the abraded CNT/epoxy composites: (**a**) measured temperature in time for the abraded area with the heat-equation model (solid lines), (**b**) heating rate and curvature (k) with a degree of abrasion ($d_{a}/d$), (**c**) convective heat ($Q_{conv,a}$) and radiative heat ($Q_{rad,a}$) at Joule heat ($P_{Joule,a}$) with steady-state temperature ($T_{steady,a}$) as a function of $d_{a}/d$, (**d**) temperature ratio with Joule heat ratio between the abraded and unabraded sections.

**Figure 7 nanomaterials-15-00337-f007:**
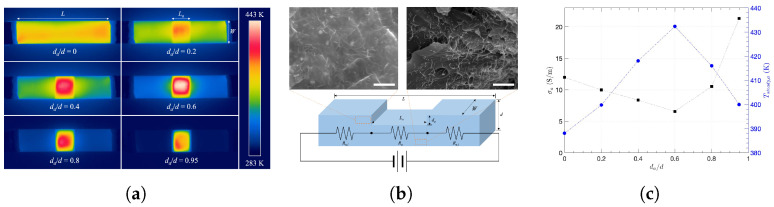
Observations of CNT/epoxy composites: (**a**) IR images of the composite with $d_{a}/d$, (**b**) the schematic of the locally abraded composite and its equivalent circuit diagram with SEM images showing CNT distribution before abrasion (upper left) and in the bottom side (upper right) of ϕ = 2.44 vol. %, both with 2 μm scale bars, (**c**) electrical conductivity ($\sigma_{a}$) and steady-state temperature ($T_{steady,a}$) of the abraded section.

**Table 1 nanomaterials-15-00337-t001:** EVA suit applications with environmental conditions and required properties [23,27].

Space	Low Temp (K)	Atmosphere	Environment	Application	Required Properties
**Earth Orbit**	153	At. O, N	Micrometeroid, Electrostatic charge	G4C orbital EVA suit for Gemini, Orbital EVA suit for shuttle/ISS EMU	Micrometeroid and thermal protective
**Moon**	25	Ar, Ne, $H_{2}$, He	Lunar dust, Electrostatic charge	A7L EVA suit for Apollo	Micrometeroid, abrasion, and thermal protective, Nonflammable
**Mars**	133	${\mathrm{CO}}_{2}$, $N_{2}$, Ar, $O_{2}$, $H_{2}$O	Martian dust, Electrostatic charge	-	-

## Data Availability

Data are contained within the article.

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
