# Peer review of "Abrasion Effect on Heating Performance of Carbon Nanotube/Epoxy Composites"

_nanomaterials, 2025, doi:10.3390/nano15050337_

Round 1

Reviewer 1 Report

Comments and Suggestions for Authors

The paper entitled “The effect of abrasion on the heating performance of carbon nanotube/epoxy composites” deals with an interesting topic with possible space applications. The authors studied the effects of abrasion on the heating performance of CNT/epoxy composites. The paper is well written, but needs minor revision before being published. I have two comments/suggestions to make:

-          the method of obtaining the studied composites, some of the characterization methods and equipment, as well as some electrical (electrical conductivity), thermal characteristics and images taken by an IR camera are presented in another form in the paper “Lee, S.-J.; Jung, Y.-J.; Cho, C.; Jang, S.-H. Effect of Atmospheric Temperature on Epoxy Coating Reinforced with Carbon Nanotubes for De-Icing on Road Systems. Nanomaterials 2023, 13, 2248.”, which is why I suggest the authors to refer to results of this paper in point 1. Introduction.

-          Figures 3-7 have poor resolution and require quality improvement.

Author Response

Response to Reviewer 1

The paper entitled “The effect of abrasion on the heating performance of carbon nanotube/epoxy composites” deals with an interesting topic with possible space applications. The authors studied the effects of abrasion on the heating performance of CNT/epoxy composites. The paper is well written, but needs minor revision before being published. I have two comments/suggestions to make:

Comments 1: The method of obtaining the studied composites, some of the characterization methods and equipment, as well as some electrical (electrical conductivity), thermal characteristics and images taken by an IR camera are presented in another form in the paper “Lee, S.-J.; Jung, Y.-J.; Cho, C.; Jang, S.-H. Effect of Atmospheric Temperature on Epoxy Coating Reinforced with Carbon Nanotubes for De-Icing on Road Systems. Nanomaterials 2023, 13, 2248.”, which is why I suggest the authors to refer to results of this paper in point 1. Introduction.

Response: Thank you for the comment. As suggested, the previous reference has been included in the introduction and characterizations.

The manuscript has revised to address the comment as follows:

At page 2;

Using the flexibility of polymers with appreciable thermal and chemical resistance, when carbon fillers are dispersed in polymeric matrix, flexible heaters have been well developed to generate Joule’s heat by the electrical current through the fillers for various applications: carbon nanotube (CNT)/epoxy composites have been considered for de-icing because epoxy resin can be integrated into structures such as bridges, terraces, roofs, and helicopter rotor blades [1]; flexible carbon fiber composites can be also used for de-icing onto the curved aircraft wings, as these are compatible with passenger aircrafts such as Airbus A350 XWB and Boeing 787, which primarily consist of carbon fiber reinforced polymer composites; flexible CNT composites have been evaluated to provide thermal comfort for passengers in vehicle seats.

At page 4;

The electrical resistance (R) of each specimen was obtained from the slope (1/R) of current-voltage (I − V ) curves by applying V from –10 V to +10 V using multimeters depending on R: Tektronix Keithley 2700 for R < 10 GΩ and Tektronix Keithley 2450 for R > 10 GΩ [1], so that its electrical conductivity (σ) was calculated as σ = L/RWd, where L is the length between the electrodes, W is the width, and d is the thickness of each specimen.

At page 4-5;

During the heating, their surface temperature distributions were monitored using a thermal infrared (IR) camera (FLIR A655sc) at 20 V, ensuring the temperature remained within the IR detector’s range [1].

[Ref]

  1. Lee, S.-J.; Jung, Y.-J.; Cho, C.; Jang, S.-H. Effect of Atmospheric Temperature on Epoxy Coating Reinforced with Carbon Nanotubes for De-Icing on Road Systems. Nanomaterials 2023, 13, 2248

Comments 2: Figures 3-7 have poor resolution and require quality improvement.

Response: Thank you for the comment. As original images have been used within latex without secondary modification, the resolutions of Figures 3-7 have been improved. 

Reviewer 2 Report

Comments and Suggestions for Authors

The work entitled Abrasion effect on heating performance of carbon nanotube/epoxy composites presents interesting results, but some points are unclear.

1 - In order to present the main findings of the study, I believe that the abstract should be improved.

2 - Regarding the concentrations of CNT used, were they based on the electrical or mechanical percolation threshold?

3 - Especially in the section on electrical percolation, the authors might find it useful to refer to the following references to enhance their discussion. In my opinion, it is very superficial. https://doi.org/10.1002/pc.24588

3a. https://doi.org/10.1016/j.compositesa.2024.108374

3b. https://doi.org/10.1039/D2SM01168A

3c. https://doi.org/10.3390/app14198973

3d. https://doi.org/10.3390/ma16083223

3e. 10.1615/CompMechComputApplIntJ.2021037544

4 - Doesn't impedance analysis contribute to a better understanding of conduction processes in conductive composites exposed to variable electric fields? Additionally, based on applications in suits and vehicles outside of the earth, wouldn't electromagnetic radiation protection testing be beneficial?

5 - What are the potential challenges associated with the scalability of CNTs for applications in space?

6 - Are CNT/epoxy composites more cost-effective and perform better than alternative materials in their intended applications?

7 - After prolonged exposure to simulated abrasion conditions, how do the materials perform?

8 - Could these composites be integrated into multifunctional systems (e.g. sensors or autonomous vehicles)?

9 - Is it possible to translate these results into applications for commercial or industrial use?

In order to submit a manuscript for review, authors must adhere to the guidelines.

Author Response

Response to Reviewer 2

The work entitled Abrasion effect on heating performance of carbon nanotube/epoxy composites presents interesting results, but some points are unclear.

1 - In order to present the main findings of the study, I believe that the abstract should be improved.

Response: Thank you for the comment. I have re-written the abstract showing the main findings of this work.

The manuscript has revised to address the comment as follows:

At page 1;

The effects of abrasion on the heating performance of carbon nanotube (CNT)/epoxy composites were investigated in terms of Joule heating, convective heat transfer, and radiative heat loss under severe and localized abrasive conditions. While the overall Joule heating behavior was characterized by the heating rate and the curvature of the transient response, a numerical solution of the heat equation was used to quantify convective and radiative heat losses, incorporating the specific heat of each component, the convective heat transfer coefficient, and the Biot number. CNT reinforcement significantly improved wear resistance at a CNT concentration of 0.31 vol. %, but the presence of micro-voids led to a slight increase in wear rate with additional CNT inclusion. While overall heating performance can be characterized as steady-state temperature, heating rate, and curvature of the corresponding heating curve. Using an equivalent circuit model, local and severe abrasion scenarios were analyzed to determine the variation in electrical conductivity with temperature at different degrees of abrasion, indicating the impact of scattering effects. This analysis provides valuable insights for estimating both wear resistance and the heating performance of self-heated surface materials, with potential applications in future space technologies.

2 - Regarding the concentrations of CNT used, were they based on the electrical or mechanical percolation threshold?

Response: Thank you for the question, the CNT concentrations were based on the electrical and thermal percolations as shown in Fig. 3(a).

The manuscript has revised to address the comment as follows:

At page 6;

Thus, using the nonlinear least square regression, Ï•c of 0.25 vol. % was experimentally obtained with the universal and material independent t of 2.0, while the effective conductivity at Ï•c is related to Ruf R1−um where Rf is the resistance of the filler, Rm is the resistance of the matrix, and u = s/(s + t). Likewise, since thermal conductivity (κ) was analyzed with the power relation, the same Ï•c of 0.25 vol. % was obtained for thermal conduction as plotted in Fig. 3 (a).

3 - Especially in the section on electrical percolation, the authors might find it useful to refer to the following references to enhance their discussion. In my opinion, it is very superficial. https://doi.org/10.1002/pc.24588

3a. https://doi.org/10.1016/j.compositesa.2024.108374

3b. https://doi.org/10.1039/D2SM01168A

3c. https://doi.org/10.3390/app14198973

3d. https://doi.org/10.3390/ma16083223

3e. 10.1615/CompMechComputApplIntJ.2021037544

Response: Thank you very much for providing the literature. I have discussed about electron hopping with the interphase layer in the section on electrical percolation.

The manuscript has revised to address the comment as follows:

At page 4;

The electrical resistance (R) of each specimen was obtained from the slope (1/R) of current-voltage (I − V) curves by applying V from –10 V to +10 V using multimeters depending on R: Tektronix Keithley 2700 for R < 10 GΩ and Tektronix Keithley 2450 for R > 10 GΩ, so that its electrical conductivity (σ) was calculated as σ = L/RWd, where L is the length between the electrodes, W is the width, and d is the thickness of each specimen [1].

At page 6;

After constructing percolation networks, as electrons can still hop across the interphase layer due to van der Waals interactions between CNTs [2], the thickness of the interphase layer would be

where dc is the maximum distance of the electron tunneling between close adjoining CNTs [3]. Considering dc = 1.8 nm [2–5], as tint = 0.4 - 0.8 nm with the obtained Ï•c of 0.25 vol. % for 0.31 ≤ Ï• ≤ 2.44 vol. %, the both percolation networks and electron hopping contribute to the effective electrical conductivity of the composites for Ï• > Ï•c [2, 6].

4 - Doesn't impedance analysis contribute to a better understanding of conduction processes in conductive composites exposed to variable electric fields? Additionally, based on applications in suits and vehicles outside of the earth, wouldn't electromagnetic radiation protection testing be beneficial?

Response: Thank you for giving me the questions. Using the Eq. (5), one can estimate Joule heating with abraded conditions of La, da, and sa under variable electric field V. As the reviewer indicated, previous studies have reported protection effect on electromagnetic radiation by CNT/polymer composites [7].

The manuscript has revised to address the comment as follows:

At page 11;

According to Eq. (5) showing the quadratic response of PJoule,a to external variable V , one can estimate higher PJoule,a at higher da with constant σa, but Fig. 7 (c) shows that σa was not consistent with da/d and it was almost doubled at da/d = 0.95 than others, although increased Ra is expected with smaller thickness of d − da.

At page 2;

While the composites might stay lightweight as it costs in the range of thousands of dollars per pound in space exploration, the tunable electrical conductivity can offer cosmic radiation shielding for protection in space and controlled temperature in response of astronaut activities from temperature induced damage using the Joule’s heat [7].

5 - What are the potential challenges associated with the scalability of CNTs for applications in space?

Response: Thank you for giving me the question. The challenges might be interaction with other severe environments in space. For example, At. O exposure reduced EMI shielding performance in low-Earth orbital space exploration [8].

The manuscript has revised to address the comment as follows:

At page 13;

Thus, this investigation provides foundation for design and analysis of CNT-polymer composites for self-heated surface materials in space applications, which are subjected to external loads or interactions with other environmental conditions, which are considered as future work, like At. O exposure in low-Earth orbits [8].

6 - Are CNT/epoxy composites more cost-effective and perform better than alternative materials in their intended applications?

Response: Thank you for giving me the question. The lightweight CNT/epoxy composites are beneficial in both cost and performance for flexible heaters in space exploration, as it costs in the range of thousands of dollars per pound [9] and the composites can act as protection and flexible heating materials.

The manuscript has revised to address the comment as follows:

While the composites might stay lightweight as it costs in the range of thousands of dollars per pound in space exploration [9], the tunable electrical conductivity can offer cosmic radiation shielding for protection in space and controlled temperature in response of astronaut activities from temperature induced damage using the Joule’s heat.

7 - After prolonged exposure to simulated abrasion conditions, how do the materials perform?

Response: Thank you for giving me the question. Unlike previous Taber wheel abrasions [10, 11], As we simulated the prolonged or severe abrasion with the sander, the heating performance was analyzed in terms of the abrased thickness da/d.

The manuscript has revised to address the comment as follows:

At page 10;

Subsequently, for even more prolonged or severe abrasions, the degree of abrasion was defined as the reduced thickness ratio da/d where d is the initial thickness and da is the abraded thickness.

At page 11;

Given applied voltage (V) to the composite, the total resistance (R) increased with less thickness (d-da) or higher degree of abrasion (d/da), so that Joule’s heat of the entire composite decreased.

8 - Could these composites be integrated into multifunctional systems (e.g. sensors or autonomous vehicles)?

Response: Thank you for giving me the question. Yes, the composites can be integrated into the multifunctional systems of sensor and heater in vehicles, as the model is based on the response of its resistance.

The manuscript has revised to address the comment as follows:

By including other effects, the method can be still used for general heater design including the multifunctional sensor and heater application in vehicles [12], and an optimal CNT concentration can be chosen to balance the required heat generation and weight loss by abrasion.

9 - Is it possible to translate these results into applications for commercial or industrial use?

Response: Thank you for giving me the question. I believe that the resistance-based method is effective to monitor status of composite materials in Boeing 787 as it has been already applied in Boeing 787.

The manuscript has revised to address the comment as follows:

By including other effects, the resistance-based method can be still used for monitoring composite status in aircrafts and for designing composite heaters including the multifunctional sensor and heater application [13, 14], and an optimal CNT concentration can be chosen to balance the required heat generation and weight loss by abrasion.

[References]

  1. Rebeque, P.V., Silva, M.J., Cena, C.R., Nagashima, H.N., Malmonge, J.A., Kanda, D.H.F.: Analysis of the electrical conduction in percolative nanocomposites based on castor-oil polyurethane with carbon black and activated carbon nanopowder. Polymer Composites 40(1), 7–15 (2019)
  2. Feng, C., Jiang, L.: Micromechanics modeling of the electrical conductivity of carbon nanotube (cnt)–polymer nanocomposites. Composites Part A: Applied Science and Manufacturing 47, 143–149 (2013)
  3. Saberi, M., Moradi, A., Ansari, R., Hassanzadeh-Aghdam, M.K., Jamali, J.: Developing an efficient analytical model for predicting the electrical conductivity of polymeric nanocomposites containing hybrid carbon nanotube/carbon black nanofillers. Composites Part A: Applied Science and Manufacturing 185, 108374 (2024)
  4. Li, C., Thostenson, E.T., Chou, T.-W.: Dominant role of tunneling resistance in the electrical conductivity of carbon nanotube–based composites. Applied Physics Letters 91(22), 223114 (2007)
  5. Takeda, T., Shindo, Y., Kuronuma, Y., Narita, F.: Modeling and characterization of the electrical conductivity of carbon nanotube-based polymer composites. Polymer 52(17), 3852–3856 (2011)
  6. Deng, F., Zheng, Q.-S.: An analytical model of effective electrical conductivity of carbon nanotube composites. Applied Physics Letters 92(7), 071902 (2008)
  7. Cha, J.-H., Jang, W.-H., Sarath Kumar, S.K., Noh, J.-E., Choi, J.-S., Kim, C.-G.: Functionalized multi-walled carbon nanotubes/hydrogen-rich benzoxazine nanocomposites for cosmic radiation shielding with enhanced mechanical properties and space environment resistance. Composites Science and Technology 228, 109634 (2022)
  8. Parkhomenko, I.N., Vlasukova, L.A., Parfimovich, I.D., Komarov, F.F., Novikov, L.S., Chernik, V.N., Zhigulin, D.V.: Atomic oxygen exposure effect on carbon nanotubes/epoxy composites for space systems. Acta Astronautica 204, 124–131 (2023)
  9. Gordon, G.D.: Space Exploration: Mass Ratios for Different Missions
  10. Gaier, J.R., Meador, M.A., Rogers, K.J., Sheehy, B.H.: Abrasion of candidate spacesuit fabrics by simulated lunar dust. Technical Report 2009-01-2473, SAE Technical Paper (2009)
  11. Ryan, E.A., Seibers, Z.D., Reynolds, J.R., Shofner, M.L.: Surface-localized chemically modified reduced graphene oxide nanocomposites as flexible conductive surfaces for space applications. ACS Applied Polymer Materials 5, 5092–5102 (2023)
  12. Park, Y., Gwon, N.-H., Seong, W.-K., Kim, W.: Heater-integrated flexible piezoresistive pressure sensor array for smart-car seats. IEEE Sensors Journal 24(2), 1255–1263 (2024)
  13. Fenta, E.W., Mebratie, B.A.: Advancements in carbon nanotube-polymer composites: Enhancing properties and applications through advanced manufacturing techniques. Heliyon 10(16), 36490 (2024)
  14. Park, Y., Gwon, N.-H., Seong, W.-K., Kim, W.: Heater-integrated flexible piezoresistive pressure sensor array for smart-car seats. IEEE Sensors Journal 24(2), 1255–1263 (2024)

Reviewer 3 Report

Comments and Suggestions for Authors

This work deals with the design of CNT/epoxy nanocomposites which can be exploited due to their abrasion effects and heating performances. The paper is interesting and it displays high scientific soundness. There are some minor revisions that should be addressed before publication:

-The abstract should be enriched with the most meaningful experimental findings.

-The graphs should be better enlightened. It is very difficult to read the numbers and the writings within each graph. Then, it is difficult to fluently read the paper.

-The porosity of the samples should be evaluated through gas adsorption and the effects of porous structure on the thermal properties should be better discussed also considering literature.

-The introduction should report the most recent advancements on the design of nanocomposite materials based also on inorganic nanotubes (such as halloysite, boron nitride, imogolite) with interesting thermal properties and related applications (phase change materials, flame retardants, etc).

Author Response

Response to Reviewer 3

This work deals with the design of CNT/epoxy nanocomposites which can be exploited due to their abrasion effects and heating performances. The paper is interesting and it displays high scientific soundness. There are some minor revisions that should be addressed before publication:

-The abstract should be enriched with the most meaningful experimental findings.

Response: Thank you for the comment. I have re-written the abstract showing the meaningful findings of this work.

The manuscript has revised to address the comment as follows:

At page 1;

The effects of abrasion on the heating performance of carbon nanotube (CNT)/epoxy composites were investigated in terms of Joule heating, convective heat transfer, and radiative heat loss under severe and localized abrasive conditions. While the overall Joule heating behavior was characterized by the heating rate and the curvature of the transient response, a numerical solution of the heat equation was used to quantify convective and radiative heat losses, incorporating the specific heat of each component, the convective heat transfer coefficient, and the Biot number. CNT reinforcement significantly improved wear resistance at a CNT concentration of 0.31 vol. %, but the presence of micro-voids led to a slight increase in wear rate with additional CNT inclusion. While overall heating performance can be characterized as steady-state temperature, heating rate, and curvature of the corresponding heating curve. Using an equivalent circuit model, local and severe abrasion scenarios were analyzed to determine the variation in electrical conductivity with temperature at different degrees of abrasion, indicating the impact of scattering effects. This analysis provides valuable insights for estimating both wear resistance and the heating performance of self-heated surface materials, with potential applications in future space technologies.

-The graphs should be better enlightened. It is very difficult to read the numbers and the writings within each graph. Then, it is difficult to fluently read the paper.

Response: Thank you for the comment. As original images have been used within latex without secondary modification, the resolutions of Figures 3-7 have been improved. 

The manuscript has revised to address the comment as follows:

-The porosity of the samples should be evaluated through gas adsorption and the effects of porous structure on the thermal properties should be better discussed also considering literature.

Response: Thank you very much for the comment. While the porosity can be evaluated by gas adsorption or using water [1, 2], we have chosen the image processing method to avoid any interaction or modification of the samples [3]. The porous effect on thermal conductivity has been discussed in terms of convective and radiative heat transfers [3].

The manuscript has revised to address the comment as follows:

At page 5;

The morphology of the specimens was examined using a scanning electron microscope (TESCAN MIRA3 FE-SEM) at 15 kV on crosss-sections of the abraded areas after Pt coating with a sputter (QUORUM Q150 TS). The images were processed to evaluate porosity without any interaction or modification of the specimens [3], while it can be also obtained by gas adsorption or using water [1, 2].

At page 10;

Moreover, as it has been considered that the higher porosity contributes to reduce k [3], Fig. 3 (a) shows that k remained almost the same although f was increased from 1.23 to 2.44 vol. %. Therefore, such excess CNT fillers not only induced more pores but also led greater weight loss of the composites during abrasion due to the reduced hardness as well as limited k.

-The introduction should report the most recent advancements on the design of nanocomposite materials based also on inorganic nanotubes (such as halloysite, boron nitride, imogolite) with interesting thermal properties and related applications (phase change materials, flame retardants, etc).

Response: Thank you very much for the comment. I have included the most recent advancements with other inorganic nanotubes for space applications in the introduction.

The manuscript has revised to address the comment as follows:

Particularly, motivated by the Artemis project, which requires to protect astronauts from harsh environments and to improve capability of mission tasks to the Moon and potential advancements for Mars [4], materials used in extravehicular activity (EVA) systems might stay lightweight as it costs in the range of thousands of dollars per pound in space exploration [5]. While other lightweight fillers, such as halloysite and boron nitride nanotubes, can provide flame retardancy and radiation shielding, respectively [6, 7], CNT-based composites can offer tunable electrical conductivity, enabling both cosmic radiation shielding and controlled temperature through Joule’s heating [8, 9].

[References]

  1. Kohlmeyer, R.R., Lor, M., Deng, J., Liu, H., Chen, J.: Preparation of stable carbon nanotube aerogels with high electrical conductivity and porosity. Carbon 49(7), 2352–2361 (2011)
  2. Molla-Abbasi, P., Ghaffarian, S.R., Danesh, E.: Porous carbon nanotube/PMMA conductive composites as a sensitive layer in vapor sensors. Smart Materials and Structures 20(10), 105012 (2011)
  3. Li, C., Fei, J., Zhang, T., Zhao, S., Qi, L.: Relationship between surface characteristics and properties of fiber-reinforced resin-based composites. Composites Part B: Engineering 249, 110422 (2023)
  4. Creech, S., Guidi, J., Elburn, D.: Artemis: An overview of nasa’s activities to return humans to the moon. In: 2022 IEEE Aerospace Conference, Big Sky, MT, USA, pp. 1–7 (2022)
  5. Gordon, G.D.: Space Exploration: Mass Ratios for Different Missions.
  6. Kang, M., Liu, Y., Lin, W., Liang, C., Cheng, J.: The thermal behavior and flame retardant performance of phase change material microcapsules with halloysite nanotube. Journal of Energy Storage 60, 106632 (2023)
  7. Cheraghi, E., Chen, S., Yeow, J.T.W.: Boron nitride-based nanomaterials for radiation shielding: A review. IEEE Nanotechnology Magazine 15(3), 8–17 (2021)
  8. Cha, J.-H., Jang, W.-H., Sarath Kumar, S.K., Noh, J.-E., Choi, J.-S., Kim, C.-G.: Functionalized multi-walled carbon nanotubes/hydrogen-rich benzoxazine nanocomposites for cosmic radiation shielding with enhanced mechanical properties and space environment resistance. Composites Science and Technology 228, 109634 (2022)
  9. Liu, C., Yin, H.: Tailorable thermoelasticity of cubic lattice-based cellular and granular materials by prestress. Materials & Design 233, 112223 (2023)
